# In Silico Study of Mangostin Compounds and Its Derivatives as Inhibitors of α-Glucosidase Enzymes for Anti-Diabetic Studies

**DOI:** 10.3390/biology11121837

**Published:** 2022-12-16

**Authors:** Ahmad Fariz Maulana, Sriwidodo Sriwidodo, Yaya Rukayadi, Iman Permana Maksum

**Affiliations:** 1Departement of Chemistry, Faculty of Mathematics and Natural Sciences, Universitas Padjadjaran, Sumedang 45363, Indonesia; 2Departement of Pharmaceutics and Pharmaceutical Technology, Faculty of Pharmacy, Universitas Padjadjaran, Sumedang 45363, Indonesia; 3Departement of Food Science, Faculty of Food Science and Technology, Universiti Putra Malaysia, Serdang 43400, Malaysia

**Keywords:** anti-diabetic, diabetes, in silico, mangostin

## Abstract

**Simple Summary:**

Diabetes is often treated with drugs, such as acarbose, metformin, and sulfonylurea. These drugs can cause various side effects such as stomach upset and gastrointestinal symptoms. Therefore, natural compounds and their derivatives, such as mangostin, are the best alternatives to these drugs. In silico potential antidiabetic tests can be performed as an initial analysis to determine which compound offers the best result. Available methods include molecular docking and molecular dynamics simulations. Based on these results, γ-mangostin had the best potential among the compounds. Therefore, it can be used to prepare better derivatives to replace acarbose as an antidiabetic drug.

**Abstract:**

Diabetes is a chronic disease with a high mortality rate worldwide and can cause other diseases such as kidney damage, narrowing of blood vessels, and heart disease. The concomitant use of drugs such as metformin, sulfonylurea, miglitol, and acarbose may cause side effects with long-term administration. Therefore, natural ingredients are the best choice, considering that their long-term side effects are not significant. One of the compounds that can be used as a candidate antidiabetic is mangostin; however, information on the molecular mechanism needs to be further analyzed through molecular docking, simulating molecular dynamics, and testing the in silico antidiabetic potential. This study focused on modeling the protein structure, molecular docking, and molecular dynamics simulations and analyses. This process produces RMSD values, free energies, and intermolecular hydrogen bonding. Based on the analysis results, all molecular dynamics simulations can occur under physiological conditions, and γ-mangostin is the best among the test compounds.

## 1. Introduction

Diabetes Mellitus (DM) is a metabolic disease caused by the lack of insulin synthesis, increased breakdown, or impaired insulin action. It is classified into two main groups, namely, type 1 and type II, and approximately 85% of diabetes cases worldwide are type 2 [1]. The number of people with type II diabetes is expected to increase rapidly in the future. Based on the IDF in 2021, Indonesia ranks 5th among countries with the highest diabetes rate worldwide, with a population of 19.5 million. This value will continue to increase, and it is estimated to increase to 28.6 million by 2045 [2].

Drug therapy, including acarbose, is the most widely used option, which is approved by the FDA for the treatment of adults with type 2 diabetes mellitus. Other treatments include diet alone or diet and exercise, depending on the patient’s health status. Acarbose adopts a competitive inhibition mechanism for α-glucosidase because its shape is similar to that of the polysaccharide chain. However, this drug has several side effects when used for a long time [3]. The most common symptoms are gastrointestinal (GI) symptoms, including flatulence, diarrhea, and abdominal pain. Monotherapy using acarbose does not cause hypoglycemia, but the use of other antidiabetic drugs, such as sulfonylurea or insulin, induces hypoglycemia [4].

Therefore, it is necessary to identify alternative compounds with relatively fewer or no side effects. Natural compounds, such as mangostin, can be used because their long-term use has significantly fewer side effects [5]. Mangostin compounds are often compared with acarbose as a positive control. However, the location of intermolecular interactions cannot be visualized, especially in the α-glucosidase enzyme, which is one of the molecular targets of acarbose [6]. This interaction was confirmed by Djeujo et al. [7] in 2022, where a colorimetric assay with increasing concentrations of α-mangostin led to reduced pNPG substrate hydrolysis. α-Glucosidase hydrolyzes pNPG into α-d-glucopyranoside and p-nitrophenol (yellow), the chromatic intensity of which decreases in proportion to the ability of an inhibitor to counteract enzyme activity. Mangostin compounds are also effective against several other target proteins such as AMPK, PPARγ, and α-amylase. Chen et al. [8] in 2021 stated that γ-mangostin is a potential target for simulating binding with this protein target. However, AMPK and PPARγ are located inside cells; AMPK is located in the nucleus and cytoplasm [9], whereas PPARγ is located in endothelial cells and vascular smooth muscle cells [10]. Because of this location, the target molecule requires further investigation for entry into the cell. This is unlike the case with the enzymes α-amylase and α-glucosidase. Both of these enzymes are a group of hydrolase enzymes that are outside the cell, also called exoenzymes, where α-amylase is located in the saliva [11] and α-glucosidase is located in the small intestine [12]. These conditions make the two enzymes more accessible as molecular targets. In addition to being inexpensive, effective, and abundant, natural compounds have been widely studied to obtain improved derivatives [13].

Although the visualization of intermolecular interactions between mangostin compounds and target proteins, such as α-glucosidase enzymes, requires further observation, various studies have proven that mangostin compounds have antidiabetic potential. In particular, those found in mangosteen have hypoglycemic and hypolipidemic activities. In vivo, their activities can repair β-pancreatic cells in streptozotocin (STZ) [14]. Baladraf [15] also demonstrated that mangostin has antihyperglycemic activity in diabetic rats induced by 40% alloxan and glucose compounds in 2021. The study was conducted using Java Plum stem and mangosteen peel extracts, as well as cinnamon. The results showed that mice administered a combination of the three extracts had lower glucose levels than those administered only one compound. This extract mixture is known to reduce glucose levels up to 76%, whereas mangosteen peel, Java Plum stem extract, and cinnamon reduced blood glucose levels by 72%, 74%, and 70%, respectively.

In vitro, acetylated mangostins, such as acarbose, have a better ability than other antidiabetic drugs. The most abundant mangostin compounds in the mangosteen extract were α-mangostin, β-mangostin, γ-mangostin, and 3-isomangostin. Based on the study conducted by Rocky et al. in 2020 [16], γ-mangostin can be acetylated to 3,6,7-trimethyl-ester-γ-mangostin, which has an IC_50_ value of 1.82 μM against the α-glucosidase enzyme. This value is higher than γ-mangostin and acarbose, which are 8.55 μM and 4.48 μM, respectively.

Furthermore, mangostin compounds can be analyzed in silico by molecular docking. The analysis results showed that the α-, β-, γ-mangostin, and sinensetin ligands interacted well with the protein maltase-glucoamylase by forming hydrogen and van der Waals bonds on the active site. The derivative with the highest affinity was α-mangostin; this is demonstrated by the binding energy value of −7.84 kcal/mol and the inhibitory constant of 1.78 M, which is lower than other mangostin derivatives (β and γ-mangostin). Sinensetin also showed a high affinity with a binding energy value of −7.45 kcal/mol and an inhibition constant of 3.46 μM. From the results of these two molecules, it can be concluded that the affinity value is relatively close compared to miglitol. This compound is often used as a positive control and has a binding energy of −9.14 kcal/mol with an inhibition constant of 1.99 μM [17].

Based on the background described above, this research used molecular docking and molecular dynamics (MD) simulations. In addition, acarbose and α-glucosidase were selected as positive controls and target proteins in this study, respectively, so that the molecular mechanisms of these enzymes could be identified.

## 2. Materials and Methods

The tools used in this study were hardware and software, including Amber20, Autodock Vina v4.2.6, AutoDock Vina 1.1.2, Biovia Discovery Studio Visualizer v21.1.0.20298, BLAST (Basic Local Alignment Search Tool) (https://blast.ncbi.nlm.nih.gov/Blast.cgi, accessed on 8 February 2022), National Center for Biotechnology Information (NCBI), Open Babel 2.4.1, PROCHECK (https://servicesn.mbi.ucla.edu/PROCHECK/, accessed on 9 February 2022), Protein Data Bank (www.rcsb.org/, accessed on 8 February 2022), PubChem (https://pubchem.ncbi.nlm.nih.gov/, accessed on 1 April 2022), and Visual Molecular Dynamics 1.9.3. The materials used were the amino acid sequence of the α-glucosidase enzyme (PDB code: 5NN4). The ligands forms consisted of α-mangostin, β-mangostin, γ-mangostin, 3-isomangostin, 1,3,6-trimethyl-ester-α-mangostin, 1,3,6,7-tetramethyl-ester-γ-mangostin, 1,6-dimethyl-ester-β-mangostin, 1,6-dimethyl-ester-3-isomangostin, and acarbose.

### 2.1. Ligand Creation

Ligands were formed using ChemDraw 15.0 and Chem3D 15.0 software, consisting of α-mangostin, 1,3,6-trimethyl-ester-α-mangostin, β-mangostin, 1,6-dimethyl-ester-β-mangostin, γ-mangostin, 1,3-dimethyl-ester-γ-mangostin, 3-isomangostin, 1,6-dimethyl-ester-3-isomangostin, and acarbose. The 2-dimensional structure of the ligand compound was formed using ChemDraw Professional 15.0. This structure was converted into a 3-dimensional version using Chem3D 15.0. It was then minimized by calculating MM2 until the RMS value was stable at 0.01 Å; this value indicates that the 3-dimensional structure is in its native state.

### 2.2. Molecular Docking

Molecular docking was carried out using AutoDock Vina, AutoDock software, and a Perl script. α-glucosidase structure was prepared using AutoDock, where the structure was cleaned from the water and the adhering ligands. Next, hydrogen was removed, polar hydrogen was re-added, and Kolmann charges were added. The grid box was then arranged on the active site of the α-glucosidase enzyme located at amino acids D616 and D518 [18]. This position is the catalytic site of α-glucosidase. The coordinates and size of the Grid Box were recorded for later use in the input file, and the prepared protein structure was saved in the PDBQT format.

The formation of the configuration document was then carried out, which contained information on the receptor file names, file ligand names, grid box size and position, number of repeated molecular docking processes, number of computer threads used, and names of log files.

The results of the molecular docking included data on the energy affinity of each ligand and the conformation of the ligand to α-glucosidase. The AutoDock software was used to evaluate the molecular docking results. The points evaluated included the binding affinity and location of the intermolecular interactions. Ligands with the correct positions were stored and visualized for intermolecular interactions using the Discovery Studio software.

Visualization with the Discovery Studio software was used to examine the types of intermolecular interactions in two dimensions (2D). The best conformation with the lowest energy value was saved in the PDB format for further molecular dynamics simulations.

### 2.3. Molecular Dynamics Simulation

MD simulations were performed using the Amber20 program for the nine complex structures that had been formed. The protein and acarbose structures were used as positive controls in the MD simulation. Before the simulation, the protein structure was separated from its ligands and inserted into the LEaP program. Subsequently, the previously separated ligands were recombined, and water molecules with an octahedral shape and Na+ ions were introduced into the system. The force fields used for the protein, ligand, and water were ff19SB, gaff2, and TIP3P, respectively. The system was minimized in three stages, involving water molecules, proteins, and the entire system, which were carried out for 2000, 2000, and 4000 cycles, respectively. Heating was performed using a weak restraint by gradually increasing the temperature from 0 °C to 37 °C. The system was then equilibrated at constant temperature and pressure for 200 ps. The production was carried out for 100 ns at constant temperature and pressure without any obstacles.

To determine the free energy values, the water molecules and Na+ ions were removed from the formed trajectory. The entire trajectory was combined to form a single file without the presence of water molecules or ions. The value of the free energy from this file was determined using the MMGBSA program, which is part of the Amber20 software.

## 3. Results

### 3.1. Structure Preparation

The structure of the α-glucosidase enzyme was obtained from the protein data bank website, which can be accessed at https://www.rcsb.org/structure/5NN4 (accessed on 8 February 2022). These structures were saved in a PDB format and cleaned by removing the ligands and any remaining water. Nine ligands were used for molecular anchoring and molecular dynamics simulations (MD).

Figure 1 shows a structure successfully formed from secondary to tertiary structures, including α-mangostin, β-mangostin, γ-mangostin, 3-isomangostin, 1,3,6-trimethyl-ester-α-mangostin, 1,3,6-tetramethyl-ester-γ-mangostin, 1,6-dimethyl-ester-β-mangostin, 1,6-dimethyl-ester-3-isomangostin, and acarbose, which served as the positive controls.

The total energy produced in the minimization process was obtained by calculating MM2, and Table 1 shows that each derivative obtained from the original compound had lower energy.

### 3.2. Molecular Docking

Molecular docking was carried out on the nine test compounds, and the box position was located at coordinates −15,671, −24.191, and 91,575, with a size of 60 × 96 × 36 and a spacing of 0.375 Å. Based on the molecular docking, the compound with the highest affinity energy was 1,6-dimethyl-ester-3-isomangostin, with an affinity energy value of −8.2 kcal/mol. This value is better than acarbose as a standard compound with an affinity energy value of −7.1 kcal/mol, as shown in Table 2.

Visualization was carried out using the Discovery Studio software, which showed the intermolecular bonds between the macromolecules and their ligands. The visualized structure was acarbose with 1,6-dimethyl-ester-3-isomangostin, as shown in Figure 2.

### 3.3. Molecular Dynamics Simulation

The simulation started from the preparation, minimization, heating, balancing, production, and analysis stages. The minimization steps up to the equilibrium produce the coordinates that will be used for the further production process.

The production step was carried out for 100 ns for all ligand structures, and the stability of the simulation system during the molecular dynamics simulation was reflected in the RMSD value. Based on the molecular dynamics simulations that have been carried out, each compound has a different RMSD value, as shown in Table 3. Acarbose, a standard compound, had an average RMSD value of 2.24 Å. The 1,3,6-trimethyl-ester-α-mangostin compound had the highest value among other mangostin compounds with a value of 2.86 Å, while the best ligand compound was the 1,6-dimethyl-ester-3-isomangostin compound with a value of 1.18 Å. The compound 1,6-dimethyl-ester-3-isomangostin was the most stable compound among other compounds.

Free energy analysis of the molecular dynamics simulation was performed using the MM/GBSA script. The results showed significant differences between the molecular docking and dynamic simulations, as shown in Table 4.

Energy component analysis using MM/GBSA showed free energy values and intermolecular interactions such as van der Waals bonds and electrostatic interactions. Based on the intermolecular interactions, the compound with the best van der Waals energy value was 1,6-dimethyl-ester-β-mangostin (−30.4346 kcal/mol) compared to the standard acarbose (−22.8293 kcal/mol). However, for electrostatic bonding, none of the eight test compounds were better than acarbose, which had a value of −61.6643 kcal/mol. γ-Mangostin had the best value among the compounds, with a value of −38.8053 kcal/mol.

The free energy formed is the accumulation of the value of ▲G_gas_ and ▲G_solv_; the γ-mangostin compound had a free energy value of −20.9956 kcal/mol, which is higher than the standard acarbose with −14.3995 kcal/mol. Intermolecular hydrogen bonds were also analyzed to determine the hydrogen bonds formed during the production process.

The results show differences for each test compound; hydrogen bonds considered significant were above 10% of the production time, as shown in Table 5.

Some test compounds did not exhibit significant intermolecular hydrogen interactions, such as α-mangostin, β-mangostin, 1,6-dimethyl-ester-β-mangostin, 3,6,7-trimethyl-ester-γ-mangostin, and 3-Isomagostin. Acarbose compounds have four intermolecular hydrogen bonds, two of which are the active sites of the α-glucosidase enzyme. Meanwhile, for the test compounds, no significant intermolecular hydrogen interactions were observed compared with acarbose.

## 4. Discussion

In silico methods can reveal the molecular mechanism of a compound, both on macromolecules and ligands. The advantages include its affordable operational costs as well as its ability to predict the results of in vitro analysis and reveal molecular mechanisms ([19,20,21,22]). Information regarding these molecular mechanisms can be used to develop diagnostic and therapeutic methods [23,24,25]. In this study, in silico examination was carried out using molecular docking, MD simulation, and analysis of α-glucosidase enzymes and their ligands. Molecular docking is performed under optimal and static conditions, whereas in the original state, the two interacting molecules are not always in this condition. Consequently, the molecular mechanism of this compound has not been fully elucidated [19]. This study emphasizes the mechanism of mangostin’s inhibition of the α-glucosidase enzyme with acarbose as a positive control in silico, using a molecular dynamics simulation. This method can reveal the interaction of a system at a spatiotemporal resolution better than other methods [21].

α-Glucosidase is often used as a therapeutic target protein for diabetes; it is an exoenzyme that works similarly to glucoamylase, which can cut glycosidic bonds in di- and oligosaccharides, as well as aryl glucosides, to produce glucose. These enzymes originate from animals, plants, bacteria, or fungi. All the plants contained α-glucosidase as an endocellular enzyme. Based on the shape of the cleaved polymer, it can only cleave the 1–4 bonds of di- or oligosaccharides and has the same function as α-amylase, but its hydrolysis rate is slower [12].

Molecular docking leads to the optimal positioning of the ligand on the macromolecule and the affinity energy value. In this study, molecular docking was performed on the α-glucosidase enzyme with acarbose and mangostin compounds as standard compounds and test compounds, respectively. Mangostin compounds have been known to inhibit α-glucosidase activity. In 2020, Djeujo et al. [7] demonstrated that a colorimetric test using pNPG with varying concentrations of α-mangostin inhibited α-glucosidase activity. This method showed that inhibition with 5 µM α-mangostin was 25.5 ± 2.6% (*p* < 0.001), while with 50 µM was 60.8 ± 2.5% (*p* < 0.001). The ability of α-mangostin compounds is in line with this research, which is based on the affinity energy values shown in Table 2. Each derivative had a value not significantly different from that of the original compound. Although the α-mangostin compound is not the best (−7.3 kcal/mol), it shows an affinity energy value that is almost similar to that of acarbose. This value indicates that the α-mangostin compound has almost the same ability as acarbose to bind the α-glucosidase enzyme. The compound with the best molecular docking value was 1,6-dimethyl-ester-3-isomangostin, with a value of 8.2 kcal/mol against the standard, namely acarbose, which has a value of −7.1 kcal/mol. Figure 2 shows that this compound can bind to amino acid ASP616 by a pi-anion interaction, while its interaction with ASP518 is only limited to van der Waals bonds. Moreover, acarbose compounds are dominated by intermolecular hydrogen bonds. They interact with amino acids ASP616 and ASP518 by intermolecular hydrogen bonds. These amino acids also bind to acarbose, although they have different types of intermolecular interactions.

There are differences between γ-mangostin and other native compounds, and the acetylated derivatives of α-, β-mangostin, and 3-isomangostin have better free energy than the original compounds. The results showed that ▲Gs_olv_ increased with an increase in ▲G_gas_. The accumulation of these two energy values becomes ▲G_total_, indicating the potential for stable bond formation. Based on the free energy values, all of these free energy values were negative, indicating that all the simulated complexes can be carried out in the laboratory. For the γ-mangostin compound, the derivative did not give a better value because the solubility of the structure was lower than that of the original compound. Given the accumulated values of ▲Gs_olv_ and ▲G_gas_, the free energy value is not as good as that of the initial compound. However, 3,6,7-trimethyl-ester-γ-mangostin had a free energy value similar to that of acarbose. The best compound from the molecular dynamics simulation results was γ-mangostin with a value of −20.9956 kcal/mol compared to the standard, namely acarbose with a value of −14.3955 kcal/mol and the α-mangostin compound with a value of −15.2123 kcal/mol. Although α-mangostin is not the best compound, it needs to be considered because, among other compounds, α-mangostin has the largest composition in mangosteen extract, which is equal to 72.04% [26] compared to γ-mangostin (6.33 %) [8]. This difference in composition is a consideration for drug manufacturing, where compounds with a higher composition will cost less in the extraction and drug manufacturing processes. Based on research conducted by Santos et al. [27] in 2022, the γ-mangostin compound had better α-glucosidase inhibitory activity than α-mangostin with a value of 11.4 ± 0.3 µM against 137 ± 2 µM, respectively. This value is in line with the MD simulation results, where the γ-mangostin compound had better free energy than α-mangostin (−15.2123 kcal/mol). This indicated that γ-mangostin is more stable than α-mangostin, making it a good candidate as an α-glucosidase inhibitor. In addition, research conducted by Chen et al. [8] showed that based on the linear regression of the in vitro test that had been carried out, the γ-mangostin compound had an IC_50_ value of 3.93 μM. The α-amylase activity of γ-mangostin was significantly higher than that of untreated controls (*p* < 0.05). γ-Mangostin also inhibited α-amylase activity in a dose-dependent manner. Furthermore, γ-mangostin showed a higher inhibition rate and lower effective concentration than acarbose (50 mg mL^−1^). This indicates that γ-mangostin has stronger anti-α-amylase and α-glucosidase activities than acarbose. These results support the findings of this study, where the γ-mangostin compound showed a better value than the other compounds, as indicated by its lower free energy value. Meanwhile, the compound with the smallest free energy is 3-isomangostin, with a value of −12.9104 kcal/mol. This result differs from that of molecular docking, which occurs under static conditions, whereas a molecular dynamics simulation occurs under dynamic conditions. This condition causes differences in the results obtained from the molecular dynamics simulations and the docking.

This intermolecular bond was strengthened by electrostatic and van der Waals bonds. Van der Waals interactions occur when adjacent atoms are close enough that their outer electron clouds barely touch each other. This action induces charge fluctuations, culminating in non-specific and non-directional attraction. This interaction is highly dependent on distance and decreases in proportion to the sixth power of the separation. The energy of each interaction is only about 4 kJ/mol, which is very weak compared to the average kinetic energy of molecules in the solution, namely, approximately 2.5 kJ/mol, and it is significant only when many interactions are combined, as in complementary surface interactions. Under the four optimal conditions, van der Waals interactions can reach bond energies as high as 40 kJ/mol or 9.56 kcal/mol [28]. In this study, the intermolecular interactions had very strong van der Waals energies. This was demonstrated by the low energy value. The compound with the lowest energy was 1,6-dimethyl-ester-β-mangostin, with a value of −30.4346 kcal/mol, compared to the standard acarbose, which has a value of −22.8293 kcal/mol.

An accurate representation of electrostatic interactions is very important in molecular dynamics simulations of biomolecular systems. Electrostatic bonds in proteins occur in ionizable amino acids such as Asp, Glu, His, Lys, and Arg [29]. In this study, the compound did not contain mangostin and its derivatives, which have a better electrostatic energy value than acarbose, with a value of −61.6643 kcal/mol. However, from all tested compounds and their derivatives, the γ-mangostin had the best value at −38.8053 kcal/mol. This difference occurs because of the different functional groups in γ-mangostin and acarbose compounds. As shown in Figure 1, acarbose has 13 hydroxyl groups, while γ-mangostin has only four hydroxyl groups. This difference causes acarbose to have a higher electrostatic energy than γ-mangostin because electrostatic bonds occur between the functional groups present in the ligand and the target protein.

Acarbose compounds have been proven to be inhibitors of α-glucosidase enzymes because they can maintain their intermolecular bonds with amino acids at the active site. There were only three compounds with intermolecular hydrogen interactions between mangostin and its derivatives. The first compound is γ-mangostin, which forms hydrogen bonds with amino acids ASP518 and ASP404. This compound is bound to one of the active sites of the α-glucosidase enzyme, ASP518. It is also capable of forming new intermolecular hydrogen bonds with the amino acid ASP404, which can stabilize its position at the site of intermolecular interactions.

Compound 1,6-dimethyl-ester-3-isomangostin forms an intermolecular hydrogen bond with amino acid TRP618, which plays a role in stabilizing the acarbose structure [18]. Although only a few ligand compounds have intermolecular hydrogen interactions, they are strengthened by van der Waals and electrostatic interactions with their respective values.

The results of this study can be extended to the stage of in vitro testing, both testing its activity and affinity for the α-glucosidase enzyme and other proteins associated with insulin secretion and insulin sensitivity, such as DPP-4 and PTP-1B [30]. Mangostin compounds can also act as antioxidants [31], which can relieve mitochondrial diabetes caused by mitochondrial DNA mutations. This mutation increases oxidative stress and causes mitochondrial diabetes. Many studies on this disease, especially bioinformatics studies, have been conducted. T10609C and C10676G mutations in mitochondrial DNA can affect proton translocation, leading to MELAS (mitochondrial myopathy, encephalopathy, lactic acidosis, and stroke-like episodes), ataxia, CPEO (chronic progressive external ophthalmoplegia), LHON (Leber hereditary optic neuropathy), and MIDD (maternally inherited diabetes and deficiency) [32]. This mutation is often accompanied by mutations in A3243G and G9053A, which can alter the structure of the mitochondrial leucine tRNA and ATPase6 [33,34] and trigger an increase in cellular oxidative stress. Therefore, mangostin compounds have the potential to become mitochondrial antidiabetic compounds that can be used in the long term.

## 5. Conclusions

Based on these results, it can be concluded that among other mangostin compounds, γ-mangostin has the best free energy value with a value of −20.9956 kcal/mol compared to the standard acarbose with a value of −14.3955 kcal/mol. Further investigations are required to determine the relationship between the in silico and in vitro experiments. Further in silico studies are needed to identify other proteins involved in diabetes.

## Figures and Tables

**Figure 1 biology-11-01837-f001:**
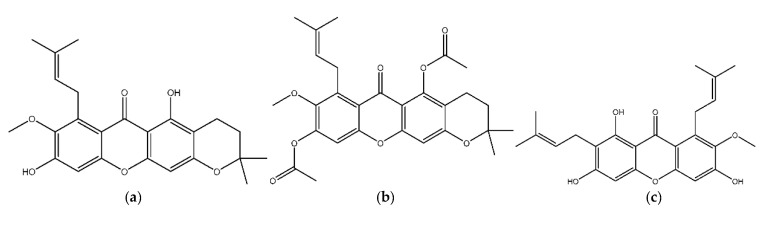
The structure of the ligand formed starts from (**a**) 3-isomangostin, (**b**) 1,6-dimethyl-ester-3-isomangostin, (**c**) α-mangostin, (**d**) 1,3,6-trimethyl-ester-α-mangostin, (**e**) β-mangostin, (**f**) 1,6-dimethyl-ester-β-mangostin, (**g**) γ-mangostin, (**h**) 3,6,7-trimethyl-ester-γ-mangostin, and (**i**) acarbose.

**Figure 2 biology-11-01837-f002:**
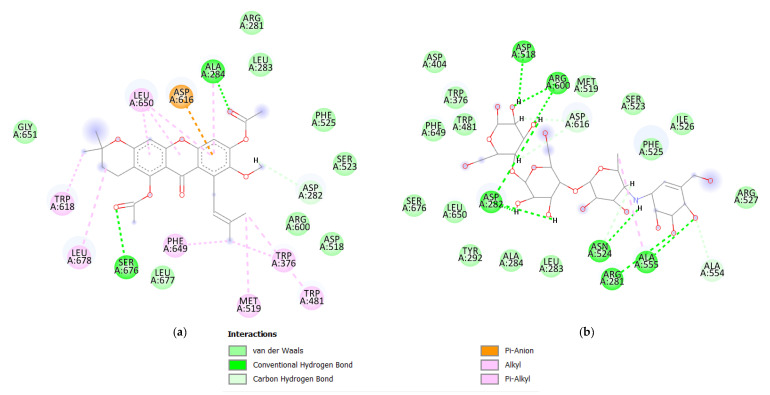
The intermolecular interaction between the compound 1,6-dimethyl-ester-3-isomangostin (**a**) and acarbose (**b**) with α-glucosidase enzyme. The 1,6-dimethy l-ester-3-isomangostin compound has a more diverse type of interaction than acarbose, especially at amino acid ASP616. 1,6-dimethyl-ester-3-isomangostin uses the pi-cation interaction to bind to the amino acid ASP616, whereas acarbose uses the Conventional Hydrogen Bond interaction to interact with ASP616.

**Table 1 biology-11-01837-t001:** The total energy value from the minimization result by calculating MM2.

No	Compound Name	Total Energy (kcal/mol)
1	α-Mangostin	25.2737
2	1,3,6-trimethyl-ester-α-mangostin	57.8246
3	β-Mangostin	32.7636
4	1,6-dimethyl-ester-β-mangostin	55.602
5	3-Isomangostin	34.796
6	1,6-dimethyl-ester-3-isomangostin	65.6732
7	γ-Mangostin	17.1138
8	1,3,6-tetramethyl-ester-γ-mangostin	54.6131
9	Acarbose	50.9353

**Table 2 biology-11-01837-t002:** Molecular docking results between the α-glucosidase protein and the nine test ligands.

Compound	Affinity (kcal/mol)
Acarbose	−7.1
α-Mangostin	−7.3
1,3,6-trimethyl-ester-α-mangostin	−7.2
β-Mangostin	−7.4
1,6-dimethyl-ester-β-mangostin	−6.7
γ-Mangostin	−7.7
3,6,7-trimethyl-ester-γ-mangostin	−7.8
3-Isomangostin	−8.1
1,6-dimethyl-ester-3-isomangostin	−8.2

**Table 3 biology-11-01837-t003:** RMSD value of the molecular dynamics simulation production stage for 100 ns. The average RMSD value was taken from 20,000 frames that were successfully formed during the production stage of the molecular dynamics simulation. The receptor is an α-glucosidase enzyme, and its ligand binds to the receptor.

No	Molecule	Receptor (Å)	Ligand (Å)
1	Acarbose	1.98	2.24
2	α-Mangostin	1.7	1.61
3	1,3,6-trimethyl-ester-α-mangostin	2.1	2.86
4	β-Mangostin	1.53	1.59
5	1,6-dimethyl-ester-β-mangostin	1.7	1.42
6	γ-Mangostin	1.7	1.34
7	3,6,7-trimethyl-ester-γ-mangostin	1.99	1.53
8	3-Isomangostin	1.75	1.38
9	1,6-dimethyl-ester-3-isomangostin	2.01	1.18

**Table 4 biology-11-01837-t004:** Energy components generated from the simulation results of molecular dynamics. All derived compounds had better free energy values than the original, but differed from mangostin, whose derivative compounds were not as good as the original.

No	Molecule	Energy Component(Kcal/mol)						
		VDWaals	EEL	EGB	Esurf	▲G (Gas)	▲G (Solv)	▲G (Total)
1	Acarbose	−22.8293	−61.6643	74.415	−4.3124	−84.4936	70.0981	−14.3955
2	α-Mangostin	−23.6659	−4.21	15.6258	−2.2922	−27.8759	12.6636	−15.2123
3	1,3,6-trimethyl-ester-α-mangostin	−26.7758	−4.7378	16.4819	−3.0714	−31.5136	13.4645	−18.0491
4	β-Mangostin	−22.6442	−2.0799	11.9138	−2.5386	−24.7241	9.3752	−15.3489
5	1,6-dimethyl-ester-β-mangostin	−30.4346	−8.7252	23.3196	−3.3416	−39.1597	19.978	−19.1817
6	γ-Mangostin	−25.2716	−38.8053	46.8243	−3.743	−64.0769	43.0813	−20.9956
7	3,6,7-trimethyl-ester-γ-mangostin	−25.2606	−17.0004	30.7017	−3.1499	−42.2611	27.5518	−14.7093
8	3-Isomangostin	−20.3336	−2.3135	12.1838	−2.4471	−22.6471	9.7367	−12.9104
9	1,6-dimethyl-ester-3-isomangostin	−25.679	−7.2242	17.9591	−2.7725	−32.9032	15.1866	−17.7166

**Table 5 biology-11-01837-t005:** Intermolecular hydrogen bonds are formed between α-glucosidase and its ligands.

No	Molecule	Hydrogen Bond
1	Acarbose	ASP518, ASP616, ASP282, ARG600
2	α-Mangostin	NA
3	1,3,6-trimethyl-ester-α-mangostin	ASN652
4	β-Mangostin	NA
5	1,6-dimethyl-ester-β-mangostin	NA
6	γ-Mangostin	ASP404, ASP518
7	3,6,7-trimethyl-ester-γ-mangostin	NA
8	3-Isomangostin	NA
9	1,6-dimethyl-ester-3-isomangostin	TRP618

NA = Not Available.

## Data Availability

Not Applicable.

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
