# Peer review of "In Silico Study of Mangostin Compounds and Its Derivatives as Inhibitors of α-Glucosidase Enzymes for Anti-Diabetic Studies"

_biology, 2022, doi:10.3390/biology11121837_

Round 1

Reviewer 1 Report

As an increase in the Diabetes mellitus prevalence is reported worldwide, authors provide data showing mangostin derivates as promising anti-diabetic drugs. Authors used in silico methods to compare 8 mangostin derivates and acarbose.

In my humble opinion, the manuscript has minor flaws which need to be adressed before acceptance:

1. The Introduction section needs extensive English revision (e.g lines 54-55, 73-74).

2. Are the results presented in lines 79-89 presented for the first time? Authors should consider English revision or moving the whole part into results section as origin of the results is not clear to the reader.

3. Authors should consider moving part of introduction (lines 90-95) into the Discussion section 

4. Authors should consider unification of compounds presented in Figure 1. In detail, only acarbose is presented as "perspective formula" while other compounds are presented as "Lewis formulas".

5. Figure 2 needs significant magnification

6. The Discussion section needs extensive English revision (e.g. lines 246-247, 256-257, 316)

7. Authors should discuss their in silico results with results of other authors (see below) already showing  effects of studied compounds (e.g. γ-Mangostin) using in vitro models.

Djeujo, Francine Medjiofack et al. “Anti-α-Glucosidase and Antiglycation Activities of α-Mangostin and New Xanthenone Derivatives: Enzymatic Kinetics and Mechanistic Insights through In Vitro Studies.” Molecules (Basel, Switzerland) vol. 27,2 547. 15 Jan. 2022, doi:10.3390/molecules27020547

Santos, Clementina M M et al. “Inhibition of the carbohydrate-hydrolyzing enzymes α-amylase and α-glucosidase by hydroxylated xanthones.” Food & function vol. 13,14 7930-7941. 18 Jul. 2022, doi:10.1039/d2fo00023g

Chen, Sih-Pei et al. “Mangosteen xanthone γ-mangostin exerts lowering blood glucose effect with potentiating insulin sensitivity through the mediation of AMPK/PPARγ.” Biomedicine & pharmacotherapy = Biomedecine & pharmacotherapie vol. 144 (2021): 112333.

8. The Results section needs  English revision (e.g. line 347)

Author Response

Thank you for taking time to review the manuscript. The following is my response to the comments that have been given:

Point 1 : The Introduction section needs extensive English revision (e.g lines 54-55, 73-74).

Response 1 : We have corrected the English in the Introduction. I also added several other references needed to add research background

Point 2 : Are the results presented in lines 79-89 presented for the first time? Authors should consider English revision or moving the whole part into results section as origin of the results is not clear to the reader.

Response 2 : The results shown in lines 78–89 are those of the study by Rocky et al. in 2020. I have clarified the results of the research, where the results of the research are listed in lines 90 - 94.

Point 3 : Authors should consider moving part of introduction (lines 90-95) into the Discussion section 

Response 3 : I've moved that section to the discussion section. actually in that section I want to explain the importance of using MD in this research, however, it needs to be re-paraphrased so it doesn't show as part of the discussion

Point 4 : Authors should consider unification of compounds presented in Figure 1. In detail, only acarbose is presented as "perspective formula" while other compounds are presented as "Lewis formulas"

Response 4 : I've converted acarbose to the Lewis formula

Point 5 : Figure 2 needs significant magnification

Response 5 : Image magnification has been performed

Point 6 : he Discussion section needs extensive English revision (e.g. lines 246-247, 256-257, 316)

Response 6 : English in the discussion section has been improved

Point 7 : Authors should discuss their in silico results with results of other authors (see below) already showing  effects of studied compounds (e.g. γ-Mangostin) using in vitro models.

Djeujo, Francine Medjiofack et al. “Anti-α-Glucosidase and Antiglycation Activities of α-Mangostin and New Xanthenone Derivatives: Enzymatic Kinetics and Mechanistic Insights through In Vitro Studies.” Molecules (Basel, Switzerland) vol. 27,2 547. 15 Jan. 2022, doi:10.3390/molecules27020547

Santos, Clementina M M et al. “Inhibition of the carbohydrate-hydrolyzing enzymes α-amylase and α-glucosidase by hydroxylated xanthones.” Food & function vol. 13,14 7930-7941. 18 Jul. 2022, doi:10.1039/d2fo00023g

Chen, Sih-Pei et al. “Mangosteen xanthone γ-mangostin exerts lowering blood glucose effect with potentiating insulin sensitivity through the mediation of AMPK/PPARγ.” Biomedicine & pharmacotherapy = Biomedecine & pharmacotherapie vol. 144 (2021): 112333.

Response 7 : For the three journals previously suggested, the authors linked the three journal results with the research results obtained by the authors in the Discussion section.

Point 8 : The Results section needs  English revision (e.g. line 347)

Response 8 : The author has improved the english writing in the results section

Author Response

Thank you for taking time to review the manuscript. The following is my response to the comments that have been given:

Point 1 : Author’s needs to cite and provide more information about Mangostin and its uses in the introduction

Response 1 : The author has added the function of mangostin in the introduction section, specifically regarding which mangostin compound is the best candidate for AGI

Point 2 : Also, why did the authors assume the mangostin interacts with α-glucosidase enzymes authors needs to provide information about this information?

Response 2 : The authors assumed that mangostin can interact with α-glucosidase based on research conducted by Chen et al. in 2021, where the study showed that the gamma-mangostin compound could inhibit the alpha-glucosidase enzyme. This inhibition certainly requires intermolecular bonds, from which the authors can assume that mangostin interacts with the α-glucosidase enzyme.

Point 3 : Did the authors try other AGI drugs to study the molecular docking? 

Response 3 : The authors also considered other AGIs such as miglitol. but based on research conducted by Mauldina et al. in 2017, showed that the IC50 value of acarbose (5.75 μg/mL) was better than miglitol (59.76 μg/mL). For this reason, the authors chose acarbose as the positive standard for alpha-glucosidase

Point 4 : γ-mangostin is shown to interact with SIRT2 enzyme with molecular docking studies and had a good sirtuin inhibitory activity. Did the authors compare the mangostin activity with different enzymes?

Response 4 : SIRT2 is closely related to tumor suppression, and our current research focuses on diabetes. However, SIRT2 should be the subject of further research. The authors also considered the activity of mangostin compounds with other enzymes, but this will be discussed in a subsequent publication.

Point 5 : Also α-glucosidase enzymes inhibitors are predominantly used to delay the carbohydrates absorption so these drugs interact with various enzymes in small intestine so how did the authors only study the interaction of mangostin with α-glucosidase enzyme?

Response 5 : For interactions with other enzymes, this will be performed in future studies. In addition, the α-glucosidase enzyme is an exoenzyme located outside the cell. Thus, it is easier for mangostin to bind to the enzyme. The selection of this enzyme was based on the ability of the standard compound (acarbose) to bind to the hydrolase enzyme. Acarbose showed better activity for alpha-glucosidase than alpha-amylase. This is our consideration when choosing the α-glucosidase enzyme to be tested.

Round 2

Reviewer 1 Report

The authors properly dealt with the comments raised and provided clear answers. Thanks authors for the additional work. The changes improved quality of the manuscript.